



# Changes in PM$_{2.5}$ concentrations and their sources in the US
# from 1990 to 2010
Ksakousti Skyllakou[1], Pablo Garcia Rivera[2], Brian Dinkelacker[2], Eleni Karnezi[2,3],
Ioannis Kioutsioukis[4], Carlos Hernandez[5], Peter J. Adams[5], and Spyros N. Pandis[1,6,*]
[1]Institute of Chemical Engineering Sciences (FORTH/ICE-HT), 26504, Patras, Greece
[2]Department of Chemical Engineering, Carnegie Mellon University, Pittsburgh, PA 15213,
USA
[3]Barcelona Supercomputing Center, 08034, Barcelona, Spain
[4]Department of Physics, University of Patras, 26500, Patras, Greece
[5]Department of Civil and Environmental Engineering, Carnegie Mellon University,
Pittsburgh, PA 15213, USA
[6]Department of Chemical Engineering, University of Patras, 26500, Patras, Greece
*Corresponding author: spyros@chemeng.upatras.gr
**Abstract**
Significant reductions of emissions of SO$_2$, NO$_x$, volatile organic compounds (VOCs)
and primary particulate matter (PM) took place in the US from 1990 to 2010. We
evaluate here our understanding of the links between these emissions changes and
corresponding changes in concentrations and health outcomes using a chemical
transport model, the Particulate Matter Comprehensive Air Quality Model with
Extensions (PMCAMx) with the Particle Source Apportionment Algorithm (PSAT).
Results for 1990, 2001 and 2010 are presented. The reductions in SO$_2$ emissions
(64%, mainly from electric generating units) during these 20 years have dominated
the reductions in PM$_{2.5}$ leading to a 45% reduction in sulfate levels. The predicted
sulfate reductions are in excellent agreement with the available measurements. Also,
the reductions in elemental carbon (EC) emissions (mainly from transportation) have
led to a 30% reduction of EC concentrations. The most important source of organic
aerosol (OA) through the years according to PMCAMx is biomass burning, followed
by biogenic secondary organic aerosol (SOA). OA from on-road transport has been
reduced by more than a factor of three. On the other hand, changes in biomass
burning OA and biogenic SOA have been modest. In 1990, about half of the US
population was exposed to annual-average PM$_{2.5}$ concentrations above 20 $\mu$g m$^{-3}$, but
by 2010 this fraction had dropped to practically zero. The predicted changes in
concentrations are evaluated against the observed changes for 1990, 2001, and 2010,





in order to understand if the model represents reasonably well the corresponding
processes caused by the changes in emissions.

**1. Introduction**
During recent decades, regulations by the US Environmental Protection Agency
(EPA) have led to significant reductions of the emissions of $SO_2$, $NO_x$, VOCs, and
primary PM from electrical utilities, industry, transportation, and other sources. Xing
et al. (2013) estimated that, from 1990 to 2010, emissions of $SO_2$ in the US were
reduced by 67%, $NO_x$ by 48%, non-methane VOCs by 49%, and primary $PM_{2.5}$ by
34%. An increase of ammonia emissions by 11% was estimated for this twenty-year
period. At the same time, there have been significant observed reductions in the
ambient $PM_{2.5}$ levels in practically all areas of the US (Meng et al., 2019). However,
our ability to link these changes in estimated emissions with the observed changes in
$PM_{2.5}$ faces challenges. The available $PM_{2.5}$ composition and mass concentration
measurements are sparse in space and are quite limited before 2001. Three-
dimensional chemical transport models (CTMs) are well suited to help address this
problem, since they simulate all the major processes that impact $PM_{2.5}$ concentrations
and transport.
There have been several efforts to quantify historical changes in $PM_{2.5}$ levels and
composition. These rely heavily on measurements (both ground and satellite for the
more recent changes) and on a number of statistical techniques including land-use
regression models to calculate the concentrations of $PM_{2.5}$ over specific areas and
periods (Eeftens et al., 2012; Beckerman et al., 2013; Ma et al., 2016; Li et al.,
2017a). More recent efforts also include applications of chemical transport models.
For example, Meng et al. (2019) estimated historical $PM_{2.5}$ concentrations over North
America from 1981 to 2016 combining the predictions of GEOS-Chem, satellite
remote sensing, and ground-based measurements. That study focused on the
estimation of total $PM_{2.5}$ levels to assess long-term changes in exposure and
associated health risks. The composition of $PM_{2.5}$ and its sources were not analyzed in
that work. Li et al. (2017a) combined in-situ and satellite observations with the global
CTM, GEOS-Chem, to quantify global and regional trends in the chemical
composition of $PM_{2.5}$ over 1989–2013. They concluded that the predicted average
trends for North America were consistent with the available measurements for $PM_{2.5}$,



secondary inorganic aerosols, organic aerosols and black carbon. Nopmongcol et al.
(2017) used CAMx with the Ozone Source Apportionment Technology (OSAT) and
Particulate Source Apportionment Technology (PSAT) algorithms for six different
years within five decades (1970-2020), to calculate the contributions from different
emission sources to $PM_{2.5}$ and $O_3$ in the US. The same meteorology and the same
natural emissions (including wildfires) were used for all six simulated years. The
authors concluded that the contribution of electrical generation units (EGUs) and on-
road sources to fine PM has declined in most areas while the contributions of sources
such as residential, commercial, and fugitive dust emissions stand out as making large
contributions to $PM_{2.5}$ that are not declining. The use of constant meteorology did not
allow the direct evaluation of these predictions.

In this study, we use period-specific meteorological data and source-resolved

emissions for every year simulated, to estimate the concentrations, composition, and
sources of $PM_{2.5}$ over 20 years in the US. The model predictions are compared with
the available measurements. The sources responsible for the $PM_{2.5}$ reductions in
various areas of the country are identified and their contribution to the reductions is
quantified. We also quantify trends in population exposure and estimated health
outcomes.

**2. Model Description**
**2.1 PMCAMx**

PMCAMx (Karydis et al., 2010; Murphy and Pandis, 2010; Tsimpidi et al., 2010;

Posner et al., 2019) uses the framework of the CAMx model (Environ, 2006) to
describe horizontal and vertical advection and diffusion, wet and dry deposition, and
gas and aqueous-phase chemistry. A 10-size section (30 nm to 40 µm) aerosol
sectional approach is used to dynamically track the evolution of the aerosol mass
distribution. The aerosol species modeled include sulfate, nitrate, ammonium, sodium,
chloride, elemental carbon, mineral dust, and primary and secondary organics. The
Carbon Bond 05 (CB5) mechanism (Yarwood et al., 2005) is used in this application
of PMCAMx for gas-phase chemistry calculations. The version of CB5 used here
includes 190 reactions of 79 surrogate gas-phase species. For condensation and
evaporation of inorganic species, a bulk equilibrium approach was used, assuming
equilibrium between the bulk inorganic aerosol and gas phases. The partitioning of





each semi-volatile inorganic species between the gas and aerosol phases is determined
by the ISORROPIA aerosol thermodynamics model (Nenes et al., 1998). The mass
transferred between the two phases in each step is distributed to the size sections
using weighting factors based on the effective surface area of each size bin (Pandis et
al., 1993). Organic aerosols (primary and secondary) are simulated using the volatility
basis set approach (Donahue et al., 2006). For primary organic aerosols (POA), 8
volatility bins, ranging from $10^{-1}$ to $10^{6}$ µg m$^{-3}$ at 298 K saturation concentration are
used. Secondary organic aerosols (SOA) are split between aerosol formed from
anthropogenic sources (aSOA) and from biogenic ones (bSOA) and modeled with 4
volatility bins (1, 10, $10^{2}$, $10^{3}$ µg m$^{-3}$) (Murphy and Pandis, 2009). NO$_x$-dependent
yields (Lane et al., 2008) are used. For better representation of the chemistry in NO$_x$
plumes, the Plume-in-Grid modeling approach of Karamchandani et al. (2011) has
been used for the major point sources following Zakoura and Pandis (2019).

**2.2 Particulate Source Apportionment Technology (PSAT)**
The PSAT algorithm (Wagstrom et al., 2008; Wagstrom and Pandis, 2011a,
2011b; Skyllakou et al., 2014; 2017) is an efficient algorithm that tracks and
computes the contributions of different sources to pollutant concentrations. The
advantages of PSAT are that it runs in parallel with PMCAMx, so there is no need to
modify the CTM for different applications and that it is quite computationally
efficient. PSAT takes advantage of the fact that the molecules of each pollutant at
each location regardless of their source have the same probability of reacting,
depositing, or getting transported to avoid repeating the simulations of these
processes. For secondary species, it follows the apportionment of their precursor
vapors. For example, the apportionment of secondary organic aerosol is based on the
apportionment of VOCs or IVOCs, sulfate on SO$_2$, nitrate on NO$_x$, and ammonium on
NH$_3$.
In this study, we use the version of PSAT developed by Skyllakou et al. (2017)
that is compatible with the Volatility Basis Set to calculate the contribution of each
emission source to the concentration of PM$_{2.5}$ and its components.





**3. Model Application**
PMCAMx-PSAT was applied over the continental United States (CONUS) for the
years 1990, 2001, and 2010 using a grid of 132 by 82 cells with horizontal dimensions
of 36 km by 36 km (covering an area of $4752 \times 2952$ km) and 14 layers of varying
thickness up to an altitude of approximately 13 km. We selected this resolution as it
has been shown to be a viable option for keeping computational and storage demands
manageable while providing sufficient quality for long-term simulations and air
quality planning applications (Gan et al., 2016). This coarse resolution introduces
errors in areas where there are significant $PM_{2.5}$ gradients in space including
California and urban areas in the rest of the western US.

**3.1    Meteorology**
Meteorological simulations were performed with the Weather Research
Forecasting model (WRF v3.6.1) over the CONUS area, with horizontal resolution of
12 x 12 km and 36 vertical (sigma) levels up to a height of about 20 km. The
simulations were executed using 3-day reinitialization from observations. Initial and
boundary conditions were generated from the ERA-Interim global climate re-analysis
database, together with the terrestrial data sets for terrain height, land-use, soil
categories, etc. from the United States Geological Survey database. The WRF
modeling system was prepared and configured in a similar way as described by
Gilliam and Pleim (2010). For the model physical parameterization, the Pleim-Xiu
Land Surface Model (Xiu and Pleim, 2002) was selected. Other important WRF
physics options used in this study include the Rapid Radiative Transfer Model/Dudhia
radiation schemes (Iacono et al., 2008), the Asymmetric Convective Model version 2
for the planetary boundary layer (Pleim, 2007a, 2007b), the Morrison double-moment
cloud microphysics scheme (Morrison et al., 2008), and version 2 of the Kain–Fritsch
cumulus parameterization (John, 2004). The selected WRF configuration is
recommended for air quality simulations (Hogrefe et al., 2015; Rogers et al., 2013).

**3.2    Emissions**
Emissions for the simulations were obtained from the internally consistent,
historical emission inventories of Xing et al. (2013) that include source-resolved gas
and primary particle emissions. Point source sectors include Electricity Generating



Units (EGU) included in the EPA's Integrated Planning Model (IPM); industrial
sources not included in the IPM (non-EGU); and all other point sources in Canada and
Mexico. Area sources include on-road emissions in the US, Canada and Mexico; off-
road emissions for the entire domain; and all remaining non-biogenic sources. We
used our WRF meteorology to drive the Model of Emissions of Gases and Aerosols
from Nature (MEGAN3) (Jiang et al., 2018) using the default emission factors for all
years to generate biogenic emissions for the CONUS domain.
In this application of PSAT, we used 6 different emission categories based on
those described above plus initial and boundary conditions which are each tracked
separately by the model as different "sources". As a result, the emission source
categories used are: 'road' which includes road emissions over the US; 'non-road'
which includes the off-road emissions of the entire domain; 'EGU'; 'non-EGU' as
described above; 'other' which includes the sum of the other point and area sources
plus the 'on-road' emissions from Canada and Mexico and finally biogenic emissions.
Figure 1 depicts the total annual emissions for each source and each year.
Biomass burning (included in the 'other' category) was the dominant source of EC
and remained relatively constant during the simulated period. The second most
important source of EC was road transport, with the corresponding emissions having
been reduced by a factor of 3.5 from 1990 to 2010. The overall reduction of EC
emissions was 40%.
Biomass burning and other sources, were the dominant source also for POA, with
almost constant contributions. Based on the emissions that Xing et al. (2013) reported
in the category 'other', we can estimate that biomass burning was responsible for 46%
of the total 'other' POA emissions. This contribution increased to 80% in 2001 and
83% in 2010. The PM emitted from biomass burning, according to the inventory, is
similar for these three years (Xing et al., 2013). The second most important source of
POA during 1990 was road transport contributing 5%. This emission source was
reduced by a factor of 3.5 from 1990 to 2010. Overall POA emissions in the inventory
were reduced by 27% from 1990 to 2010.
Emissions of VOCs by on-road sources were reduced by a factor of 3.5 during
these 20 years. On the other hand, the VOCs emitted by non-road transport decreased
by only 8%. The biogenic VOC emissions varied from year to year based on the



prevailing meteorology, but their changes were less than 20%. The total
(anthropogenic and biogenic) VOC emissions decreased by 31% from 1990 to 2010.
The emissions of the most important $SO_2$ source, EGUs, were reduced 33% from
1990 to 2001 and 67% from 1990 to 2010. This resulted in a 64% reduction of the
total $SO_2$ emissions over these 20 years.
For $NH_3$, the most important source is agriculture (included in the 'other'
category), and the corresponding emissions increased by 9% during these 20 years.
Road transportation is one of the major $NO_x$ sources with the corresponding
emissions having been reduced by 21% from 1990 to 2001 and 58% from 1990 to
2010. The second most important source for $NO_x$ in 1990, were the EGUs, which
emitted 25% less $NO_x$ in 2001 and 66% less in 2010 compared to 1990. Total $NO_x$
emissions in the inventory were 47% lower in 2010 compared to 1990.

**4. Results**
**4.1 Annual-average concentrations and sources**
We examine first the source apportionment results of PMCAMx-PSAT for the
major components of $PM_{2.5}$ for the three simulated years.
On-road transportation was a major source of EC especially in urban areas in 1990
(Figure 2). The EC concentrations originating from this source were reduced by more
than a factor of 3 from 1990 to 2010. The industrial sources (EGUs and non-EGU)
contributed less than 0.1 µg m$^{-3}$ of EC in all areas during these years. The 'other'
source which includes all types of biomass burning was the most important source
during the simulated period. Long range transport (LRT), which represents the
transport from areas outside of the domain, contributed approximately 0.1 µg m$^{-3}$.
The predicted average total OA levels defined as the sum of POA and SOA are
shown in Figure 3. The OA originating from road transport was about 0.7 µg m$^{-3}$
during 1990 over the Eastern US, but it was reduced to less than 0.5 µg m$^{-3}$ during
2010. 'Non-road' transport and 'non-EGU' emission sources had smaller
contributions to OA, with less than 0.2 µg m$^{-3}$ in most areas during all years. Biogenic
SOA was almost 1 µg m$^{-3}$ over the south-east US both during 1990 and 2001, but
during 2010 it had higher concentrations in some areas. Especially in the South due to
local meteorology predicted SOA was much higher compared to 1990. In 2010, the
biogenic VOC concentrations were on average 15% higher compared to 1990 due





mainly to the meteorological conditions during these two specific years. This small
increase is consistent with the biogenic VOC emissions estimated by Sindelarova et
al. (2014). Also, high OA concentrations were predicted to originate from biomass
burning during 1990. The average contribution of long-range transport OA was
approximately 0.6 µg m$^{-3}$.
Sulfate was the dominant component of PM$_{2.5}$ in the Eastern US in 1990 and the
EGUs were its dominant source contributing more than 5 µg m$^{-3}$ over wide areas of
the East (Figure 4). The corresponding sulfate concentrations from EGUs were
reduced to 3 µg m$^{-3}$ in 2001 and to 1.5 µg m$^{-3}$ in 2010 due to the dramatic reduction of
these SO$_2$ emissions over these 20 years. Sulfate concentrations originating from non-
EGU and other emission sources were 1 µg m$^{-3}$ or less during all years. Long-range
transport contributed approximately 0.9 µg m$^{-3}$ to the sulfate levels during the
simulated period.

**4.2 Regional contributions of sources to PM$_{2.5}$ components**
The US was divided in seven regions (Fig. 5) to facilitate the spatial analysis of
the source contributions and their changes during the simulated period. The Northeast
(NE) region includes major cities such as New York, Boston, Philadelphia, Baltimore
and Pittsburgh, while the Mideast (ME) includes the Ohio-river valley area with a
number of electrical generation units. The Midwest (MW) has significant agricultural
activities, while much of the West (WE) is relatively sparsely populated. California
(CA) was kept separate from the other western regions. The southern US was split
into a southeast region (SE) with significant biogenic emissions and the southwest
(SW) with much less vegetation.
Figure 6a shows the predicted average concentrations of EC for each year in each
region. The highest concentrations for 1990 were predicted in Northeast, followed by
the Mideast and the California. Biomass burning, included in the 'other' source, was
the dominant source of EC in all regions, with relatively constant concentration
through the years, except from CA, where the contribution from this source in 1990
was much higher due to the annual variation in fires. There was significant reduction
of the EC levels in all regions except for the West, where the EC originates mainly
from biomass burning and long-range transport. The highest reductions were
predicted for the eastern US. Figure 6b shows the population exposure (Walker et al.,





1999), which is calculated in this work as the product of the average annual
concentration of each computational cell times the population living in the cell. The
US population distribution was calculated for each year based on the US Census
Bureau (2019) data. The population exposure is significant in areas with high
population density, for example in CA.

The source contributions to the annual-average concentrations of OA are depicted

in Figure 7a. The predicted concentrations of OA in 1990 in the eastern US (NE, SE
and ME regions) were almost 3 μg m$^{-3}$ and in the other regions, less than 2.5 μg m$^{-3}$.
OA originating from biomass burning dominated the concentrations of OA during all
years and regions. Biogenic SOA was the second most significant OA component in
the Southeast. OA originating from on-road transport contributed, according to the
model, almost 0.5 μg m$^{-3}$ during 1990 and almost 0.2 μg m$^{-3}$ during 2010 in the
eastern US. Significant reductions of OA are predicted for the Northeast, Mideast, and
California while moderate reductions for the Midwest, West, and Southwest. The OA
in the Southeast has more complex behavior due to the predicted increase of biogenic
SOA in 2010 that leads to a small increase of the total OA compared to 2001. The
population exposure for OA (Figure 7b) is almost the same for Northeast and Mideast
during 1990 and it decreased during 2001 and 2010. For the Midwest, West, and
Southwest the population exposure to OA remained almost constant though the years.
For all regions, the highest population exposure was due to biomass burning and the
"other" sources. In addition, 20% of the population exposure was due to road
transport during 1990 at the highest populated areas (NE, ME, and CA), but this
percentage was reduced to almost 10% during 2010.

The highest concentrations of sulfate for 1990 are predicted in the Eastern US

(NE, ME and SE) in regions downwind of the EGUs which are the dominant SO$_2$
source in these areas (Fig. 8a). The drastic reductions of the EGU emissions are
predicted to have led to major reductions in the sulfate levels in these three regions.
More modest, but significant reductions of sulfate are also predicted for the Midwest
and the Southwest. The reductions in the West and in California from the EGU source
are small given that the sulfate there even in the 1990s was relatively low and was
dominated on average by long-range transport. Regarding the population exposure for
NE and ME, the percentage of population exposure due to EGUs during 1990 was





58% for the NE and 64% for the ME, but during 2010 these percentages were reduced
to 44% and 53% respectively.
The mortality rates caused by total $PM_{2.5}$ were also calculated for the three
simulated periods, following the relationships of Tessum et al. (2019) and using the
death rates of US population by Murphy et al. (2013). We estimated 861 deaths per
100,000 persons for 1990, 777 for 2001, and 658 for 2010.

**4.3 Linking average changes in emissions, concentrations, and exposure**
The 72% reduction of emissions of EC from road transport, from 1990 to 2010
according to PMCAMx led to a 72% reduction of EC concentrations and a 70%
reduction in human exposure to EC from this source (Table 1). The changes in
concentrations are practically the same as those of the emissions because EC is inert
and the atmospheric processes that affect it (transport and removal) are close to linear.
The small difference between the change in emissions and that of exposure is due to
small differences in the spatial distributions of the EC concentrations from road
transport and the population density. The differences are small because most road
transport emissions are in densely populated areas. The similarity in the fractional
change of emissions and concentrations applies as expected to all EC source types
(Table 1). However, for all these other sources the reduction in exposure is less than
the reduction in emissions (or concentrations). For example, the 44% reduction of EC
emissions from non-road transport, was accompanied by a 43% reduction in
concentrations, but a 35% reduction of human exposure. This is due to the location of
the reductions of these non-road transport emissions. A significant fraction of these
reductions took place away from densely populated regions (e.g., in agricultural
regions) therefore they resulted in a smaller reduction of human exposure. The
situation is a little different for total EC. The 40% reduction in emissions is predicted
to have led to a 31% reduction in concentration. The difference here is due to the
contribution of long-range transport (sources outside of the US) which are assumed to
have remained approximately constant during this period. The predicted reduction in
exposure is 33% and is due to the local sources. The changes in EC exposure in each
region are depicted in Figure 6b.
The changes in fresh POA are a little more interesting, because it is treated as
semi-volatile and reactive in PMCAMx. For all US sources, the reduction in



concentrations is a little higher than that of the emissions (Table 1). For example, a
25% reduction of POA emissions of non-road POA, is predicted to have resulted in a
30% reduction of the POA concentrations. This difference is due mostly to the non-
linear nature of the partitioning of these emissions between the gas and the particulate
phase. As the emissions are reduced, the corresponding OA concentrations are
reduced and more of the organic material is transferred to the gas phase to maintain
equilibrium. This additional evaporation leads to an additional reduction of the POA
concentrations. This is the case for all sources, so the 27% reduction in POA
emissions corresponds according to PMCAMx to a 33% average reduction in POA
concentrations. The reduction in exposure is, in absolute terms, a little less than that
of the concentrations for the same reasons as for EC. This difference is small (-74%
versus -71%) for road transport, but more significant for sources located outside urban
centers (e.g. for EGU it is -13% for concentrations and -6% for exposure).

The reductions predicted by PMCAMx for SOA (aSOA+bSOA) concentrations

are far more complex than those of fresh POA, since the formation of secondary
organic species involves non-linear processes such as partitioning, dependence on
oxidant levels, $NO_x$-dependence of the yields, and the complexity of the chemical
aging. Overall, PMCAMx predicts that the reductions in exposure are less than the
reductions in average concentrations over the US which are also less than the
reductions in the emissions of the anthropogenic volatile and intermediate volatility
organic compounds. The major reasons for this behavior are the simultaneous
decreases in $NO_x$ that have led to increased SOA formation yields and the time
required for the formation of this SOA which is often produced away from its sources
in high urban density areas. The reasons for this behavior are complex and will be
analyzed in detail in future work. However, at least part of the explanation is due to
decreases in $NO_x$ concentrations over the same period and associated increases in
SOA yields.

The predicted reductions in sulfate concentrations are less than the reductions in

emissions due mainly to the non-linearity of the aqueous-phase conversion of $SO_2$ to
sulfate (Seinfeld and Pandis 2016) (Table 1). Taking into account the transport of
some of the sulfate from areas outside of the US, the model predicts that the 64%
reduction in $SO_2$ emissions has resulted in a 45% reduction of the sulfate





concentration on average. The change in exposure is a little less, -40% on average due
mainly to the location of the major $SO_2$ sources relatively away from urban centers.

**4.4 Distribution of population exposure to $PM_{2.5}$ from different sources**
We have calculated the percentage of people exposed to different $PM_{2.5}$
concentrations from the major sources ('other', 'EGUs', 'road transport') for the three
different periods. Almost half of the US population was exposed to $PM_{2.5}$
concentrations above 20 μg m$^{-3}$ in 1990. A decade later this percentage was less than
20% and close to zero during 2010 (Fig. 9a). During 1990, almost 90% of the US
population was exposed to $PM_{2.5}$ concentrations above 10 μg m$^{-3}$, the suggested
annual mean by the World Health Organization (WHO, 2006). This percentage was
reduced to 83% in 2001 and 70% in 2010 (Fig. 9a and Fig. S2h).
The predicted distribution of the population exposed to $PM_{2.5}$ from the source
'other' in 1990 covered a wide range extending from approximately 1 to 16 μg m$^{-3}$.
The exposure from these sources was reduced significantly in the following years
mainly due to the reductions in the emissions of paved/unpaved road dust, prescribed
burning, and industrial emissions (Xing et al., 2013). The average emissions from
wildfires did not change appreciably, but this distribution was sharper in 2010, with
maximum percentages of people exposed appearing for $PM_{2.5}$ concentrations ranging
from 5 to 8 μg m$^{-3}$. The random spatial variation of biomass burning sources can
affect areas with different population density.
The exposure of the population to primary and secondary $PM_{2.5}$ from EGUs has
been dramatically decreased (Fig. 9c). In 1990 according to PMCAMx 56% of the US
population was exposed to more than 3 μg m$^{-3}$ from this source. This percentage was
reduced to 39% in 2001 and to 2% in 2010. For the threshold of 5 μg m$^{-3}$ the
reduction was from 18% in 1990, to 1% in 2001 to practically zero in 2010.
Similarly, significant decreases are predicted for road transport $PM_{2.5}$. While in
1990, 79% of the population was exposed to levels exceeding 1 μg m$^{-3}$, this
percentage was 58% in 2001 and 18% in 2010 (Fig. 9d). The corresponding changes
for the 2 μg m$^{-3}$ were from 27% (1990) to 8% (2001) to zero (2010).







**4.5 Evaluation of the model**
The model was evaluated on annual basis against ground level measurements
from the IMPROVE and CSN networks (STN U.S. EPA, 2002; IMPROVE, 1995).
The metrics used (Fountoukis et al., 2011), include the normalized mean bias (NMB),
the normalized mean error (NME), the mean bias (MB), the mean absolute gross error
(MAGE), the fractional bias (FBIAS), and the fractional error (FERROR),

$$NMB = \sum_{i=1}^{n}(P_i - O_i)\Big/\sum_{i=1}^{n}O_i \qquad NME = \sum_{i=1}^{n}|P_i - O_i|\Big/\sum_{i=1}^{n}O_i$$
$$MB = \frac{1}{n}\sum_{i=1}^{n}(P_i - O_i) \qquad MAGE = \frac{1}{n}\sum_{i=1}^{n}|P_i - O_i|$$
$$FBIAS = \frac{2}{n}\sum_{i=1}^{n}(P_i - O_i)\Big/(P_i + O_i) \qquad FERROR = \frac{2}{n}\sum_{i=1}^{n}|P_i - O_i|\Big/(P_i + O_i)$$
where $P_i$ represents the model-predicted value for site $i$, $O_i$ is the corresponding
observed value and $n$ is the total number of sites. During 1990, there were only 27
measurement sites available from the IMPROVE network, but this number increased
dramatically in 2001 and 2010. The results are summarized in Table 2.
We have excluded the region of California from this analysis because the coarse
resolution used in this application does not allow PMCAMx to capture the significant
gradients and high concentrations observed in that area.
According to Morris et al. (2005), the level of the performance of the model
would be considered as excellent if it meets the following criteria: FBIAS ≤ ± 15%
and FERROR ≤ 35%; and is good if FBIAS ≤ ± 30% and FERROR ≤ 50%. Based
on the above criteria the model performance is excellent for annual averages of PM$_{2.5}$,
OA, and ammonium for all years. The EC performance is good to excellent, while
that for sulfate it is excellent of 1990 and 2010 but average for 2001. The current
version of PMCAMx has difficulties reproducing the nitrate levels with performance
that varies from average to good. There several reasons for these problems including
the spatial resolution used here, the assumption of bulk equilibrium, etc., that will be
analyzed further in future work. PMCAMx has a small tendency towards
overprediction of the OA and underprediction of the EC. There is also a tendency
towards overprediction of the sulfate and as a result, the ammonium too. The



fractional error for nitrate is closer to 0.5 with the model in general underpredicting
the observed values.
The predictions for $PM_{2.5}$ concentrations, for which there are many more stations
and thus available measurements in 2001 and 2010, are reproduced with fractional
bias of 9% and fractional error less than 25%. For 1990, there is little bias, while there
is a small tendency towards overprediction in the later years. The performance of the
model differs with the region examined (Table S1), for example the model tends to
underpredict $PM_{2.5}$ and its components in the West part of US. The coarse resolution
used here is not sufficient to represent the gradients observed between some relatively
isolated urban areas and the relatively clean background.
One of the important results of this evaluation is the relatively consistent
performance of PMCAMx during the different years. The use of a consistent emission
inventory, consistent meteorology and measurements have probably contributed to
this outcome.

**4.5.1 Predicted spatial changes of concentrations**
We calculated the predicted changes in annual-average concentrations between
1990 and 2010 for the main $PM_{2.5}$ components. Figure S3 shows the reductions in EC
concentrations from 1990 to 2010. The reductions of the EC emissions resulted in
total reductions of the average concentrations of around 30% in the twenty-year
period. Reductions above 20% are predicted not only in the large urban areas but also
in large regions in both the eastern and the western US.
Average organic aerosol levels were reduced according to PMCAMx by close to
1.5 $\mu g\ m^{-3}$ from 1990 to 2010 in a wide area extending from the Great Lakes to
Tennessee, but also in parts of the Eastern seaboard (Fig. S4). These reductions
correspond to 35-45% of the OA in both the Northeast and California.
From 1990 to 2010, sulfate was reduced by 50-60% in the part of the country to
the east of the Mississippi. The corresponding reductions in the middle of the country
and in the western states from 1990 to 2010 were in the 20-30% range for the
relatively low sulfate levels in these regions (Fig. S5). These simulations suggest that
the Eastern US has benefited more both in an absolute and in a relative sense from
these reductions in $SO_2$ emissions.





458   We also compared the predicted and observed concentration changes, using the

459   Pearson's correlation coefficient and the average percentage differences, summarized

460   in Table 3. For the first two cases (1990 to 2001 and 1990 to 2010) there were only a

461   few measurements available for 1990. The model reproduces quite well the predicted

462   changes against the observed for $PM_{2.5}$ and its components (Fig. S6).

463   For EC, the correlation was high between 1990 and 2001, with r = 0.80; and

464   between 1990 and 2010, with r = 0.91. However, the analysis for the changes up to

465   2010 is complicated by the change in the EC measurement protocol in several CSN

466   sites in the period from 2007 to 2010. The change from the Thermal Optical

467   Transmittance (TOT) to the Thermal Optical reflectance (TOR) resulted in small

468   increases in the reported EC that were of similar magnitude as the predicted changes

469   due to the emissions reductions. To partially address this issue, we do not include in

470   the analysis the results from 14 CSN sites which reported increases in the EC from

471   2001 to 2010. Excluding these sites an r = 0.39 is calculated (Table 3). The data

472   points from these sites can be seen in the lower triangle of Fig. S6. The reduced r for

473   the 2001-10 is probably due, at least partially, to this uncertainty of the measured

474   changes.

475   The predicted average change of OA in the measurement sites from 1990 to 2001

476   was -13%, in good agreement with the observed -16% in the same locations. The

477   predicted changes were reasonably well correlated (r = 0.68) with the measured ones

478   during this decade. However, the model performance during the next decade (2001-

479   10) deteriorates as it underpredicts on average the changes (predicted -9% versus

480   observed -18%) and the changes are not correlated to each other in space. Additional

481   analysis suggested that, while the model does a reasonable job reproducing the

482   changes in the western half of the country and the northeastern quarter, it overpredicts

483   the OA concentration in 2010 and thus underpredicts the reductions in the

484   southeastern US. Our analysis also suggests that this mainly due to an overprediction

485   of the biogenic SOA in this part of the country. This is consistent with the anomalous

486   predicted increase of biogenic SOA from 2001 to 2010 in the SE US (Figure 7 and

487   Figure S1). This interesting discrepancy regarding the predicted and observed changes

488   of biogenic SOA will be analyzed in detail in a subsequent paper.

489   For sulfate, the model reproduced well the observed changes for the three

490   comparison periods, with Pearson's correlation coefficient r = 0.88 (from 1990 to





2001); 0.97, from 1990 to 2010; and 0.92, from 2001 to 2010 (Table 3). Despite the
nonlinearity in the behavior of sulfate, the average predicted and observed percentage
changes were consistent for the three comparison periods.
Finally, for PM$_{2.5}$ the model reproduces well the observed changes for the three
comparison periods with r = 0.81 (from 1990 to 2001); 0.82 (from 1990 to 2010) and
0.61 (from 2001 to 2010). The average percentage changes for the observations and
the predictions were close for all the cases (Table 3).

**5. Conclusions**
The CTM, PMCAMx, was used to simulate the changes in source contributions to
PM$_{2.5}$ and its components over two decades accounting for changes in emissions and
meteorology with internally consistent methods. Biomass burning and 'other' sources,
primarily including construction processes; mining; agriculture; waste disposal, and
other miscellaneous sources, contributed approximately half of the total (primary and
secondary) PM$_{2.5}$ during the examined 20-year period. The corresponding average
PM$_{2.5}$ concentration levels due to this group of sources have been reduced by 33%
from 1990 to 2010. EGUs were the second most important source of PM$_{2.5}$; the
corresponding ambient PM$_{2.5}$ levels have been reduced by 55% and their contribution
to the total from 16% to 11%. On-road transport was the third most important source
of PM$_{2.5}$. The total average PM$_{2.5}$ from this source was reduced by 59%, while their
contribution to the average PM$_{2.5}$ levels has been reduced from 8% to 5%.
OA was a significant fraction of PM$_{2.5}$. Biomass burning included in the 'other'
sources was the most important source of OA with fractional contributions varying
from 38% to 52% depending on the region. Biogenic SOA was the second dominant
component of OA with contributions ranging from 6% to 22% in the South US.
The reduction in exposure was less than the reduction in emissions (or
concentrations) for sources that are located away from densely populated regions
(non-road transport and non-EGUs) due to the spatial non-uniformity of the
corresponding PM$_{2.5}$ reductions. For example, sulfate human exposure by non-EGU
source was reduced by 46% from 1990 to 2010, while the corresponding reduction in
emissions was 62%.
From 1990 to 2010, the reduction of human exposure to EC was 33%, to fresh
POA 35%, to sulfate 40%, and to SOA (both anthropogenic and biogenic) 8%. The



reduction of EC was mostly due to the 72% reduction of on-road EC emissions, while
the reduction in sulfate to the 64% reduction of $SO_2$ emissions from EGUs.

During the 20 year-long examined period, the fraction of the US population

exposed to average $PM_{2.5}$ concentrations above 20 µg m$^{-3}$ decreased from
approximately 50% to close to zero. In 1990, 12% of the US population was exposed
to $PM_{2.5}$ concentrations lower than the suggested annual mean by the WHO (10 µg
m$^{-3}$). This fraction increased to 30% in 2010.

PMCAMx reproduced the annual average concentrations of $PM_{2.5}$ with fractional

error less than 30% for the three simulation periods. The corresponding fractional
biases were 19% for 1990 and 9% for both 2001 and 2010. The model also reproduces
well the average reduction of $PM_{2.5}$ in the measurement sites; the measured reduction
was 28% while the model predicts a reduction of 30%. A model weakness that
requires additional investigation is its tendency to predict an increase in the biogenic
SOA from 2001 to 2010 that appears inconsistent with the observations.

**6. Code and data availability**

The code and simulation results are available upon request

(spyros@chemeng.upatras.gr).

**7. Supplement**

**8. Author contributions**

546        K.S performed the PMCAMx and PSAT simulations, analyzed the results and

wrote the manuscript. P.G.R. prepared the anthropogenic emissions and other inputs
for the simulations. B.D. performed MEGAN simulations and analyzed the results;
E.K. performed and evaluated the WRF simulations; I.K. set-up the WRF simulations
and assisted in the preparation of the meteorological inputs. C.H. analyzed the
simulation output. S.N.P. and P.J.A. designed and coordinated the study and helped in
the writing of the paper. All authors reviewed and commented on the manuscript.

**9. Competing interests**

The authors declare that they have no conflict of interest.






## 10. Acknowledgments

This work was supported by the Center for Air, Climate, and Energy Solutions (CACES) which was supported under Assistance Agreement No. R835873 awarded by the U.S. Environmental Protection Agency.

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

Contributions of local and regional sources to fine PM in the megacity of Paris,
Atmos. Chem. Phys., 14, 2343–2352, 2014.

Skyllakou, K., Fountoukis, C., Charalampidis, P., and Pandis, S. N.: Volatility-
resolved source apportionment of primary and secondary organic aerosol over
Europe, Atmos. Environ., 167, 1–10, 2017.

Tessum, C., W., Apte, J., S., Goodkind, A., L., Muller, N., Z., Mullins, K., A.,
Paolella, D., A., Polasky, S., Springer, N., P., Thakrar, S., K., Marshall, J., D., Hill,
677       J., D.: Inequity in consumption of goods and services adds to racial–ethnic
disparities in air pollution exposure, Proc. Natl. Acad. Sci. 116, 6001–6006, 2019.

Tsimpidi, A. P., Karydis, V. A., Zavala, M., Lei, W., Molina, L. T., Ulbrich, I. M.,
Jimenez, J. L., Pandis, S. N., Evaluation of the volatility basis-set approach for the
simulation of organic aerosol formation in the Mexico City metropolitan area,
Atmos. Chem. Phys., 10, 525-546, 2010.

US CENSUS Bureau: https://www.census.gov, last access: November 2020.



U.S. Environmental Protection Agency: User Guide: Air Quality System, Report,
Research Triangle Park, N. C., Apr. Available at: www.epa.gov/ttn/airs/airsaqs/
manuals/AQSUserGuide.pdf, 2002 (last access: September 2020).

Wagstrom, K. M., Pandis, S. N., Yarwood, G., Wilson, G. M., Morris, R. E.:
Development and application of a computationally efficient particulate matter
apportionment algorithm in a three-dimensional chemical transport model, Atmos.
Environ., 42, 5650-5659, 2008.

Wagstrom, K. M., Pandis, S. N.: Source receptor relationships for fine particulate
matter concentrations in the Eastern United States, Atmos. Environ., 45, 347-356,
2011a.

Wagstrom, K. M., Pandis, S. N.: Contribution of long-range transport to local fine
particulate matter concerns, Atmos. Environ., 45, 2730e2735, 2011b.

WHO, Air Quality Guidelines for Particulate Matter, Ozone, Nitrogen Dioxide and
Sulfur Dioxide, GLOBAL Update 2005, Summary of Risk Assessment, World
Health Organization (WHO/SDE/PHE/OEH/06.02), 2006.

Xing, J., Pleim, J., Mathur, R., Pouliot, G., Hogrefe, C., Gan, C. M., Wei, C.:
Historical gaseous and primary aerosol emissions in the United States from 1990 to
2010, Atmos. Chem. and Phys., 13, 7531–7549, 2013.

Xiu, A., Pleim, J. E.: Development of a land surface model. Part I: Application in a
Mesoscale Meteorological Model, J. of Applied Meteorology, 40, 192–209, 2002.

Walker, S.E., Slordal, L. H., Guerreiro, C., Gram, F., Gronskei, K. E.: Air pollution
exposure monitoring and estimation part II. Model evaluation and population
exposure, J. Environ. Monit., 1, 321–326, 1999.

Yarwood, G., Rao, S., Yocke, M., Whitten, G. Z.: Updates to The Carbon Bond
Chemical Mechanism, Research Triangle Park, 2005.

Zakoura, M., and Pandis, S. N: Improving fine aerosol nitrate predictions using a
Plume-in-Grid modeling approach, Atmos. Environ., 215, 116887, doi: 10.1016/
j.atmosenv.2019.116887, 2019.




**Table 1**: Percentage changes in emissions from each source, and corresponding
changes in average concentrations and exposure from 1990 to 2010.

| | Road | Non-road | EGU | Non-EGU | Biogenic | Other | Total |
|---|---|---|---|---|---|---|---|
| **EC** | | | | | | | |
| **1990 to 2010** | | | | | | | |
| **Emissions (EC)** | -72 | -44 | -13 | -7 | - | -17 | -40 |
| **Concentrations** | -72 | -43 | -13 | -8 | - | -18 | -31 |
| **Exposure** | -70 | -35 | -3 | 4 | - | -12 | -33 |
| **Fresh POA** | | | | | | | |
| **1990 to 2010** | | | | | | | |
| **Emissions (fresh POA)** | -72 | -25 | -13 | -14 | - | -25 | -27 |
| **Concentrations** | -74 | -30 | -13 | -20 | - | -31 | -33 |
| **Exposure** | -71 | -25 | -6 | -11 | - | -32 | -35 |
| **SOA** | | | | | | | |
| **1990 to 2010** | | | | | | | |
| **Emissions (IVOCs+VOCs)** | -71 | -8 | -8 | -31 | 15 | -34 | -31 |
| **Concentrations** | -71 | -17 | -11 | -21 | 23 | -27 | -21 |
| **Exposure** | -66 | -6 | 1 | -8 | 37 | -18 | -8 |
| **Sulfate** | | | | | | | |
| **1990 to 2010** | | | | | | | |
| **Emissions (SO$_2$)** | -93 | -51 | -67 | -62 | - | -52 | -64 |
| **Concentrations** | -91 | -44 | -63 | -54 | - | -38 | -45 |
| **Exposure** | -88 | -30 | -60 | -46 | - | -27 | -40 |





**Table 2**: Evaluation metrics for annual average concentrations of PM$_{2.5}$ and for its
major components for each examined year.

| | MB (µg m$^{-3}$) | MAGE (µg m$^{-3}$) | NMB | NME | FBIAS | FERROR | Stations | Comment |
|---|---|---|---|---|---|---|---|---|
| **EC** | | | | | | | | |
| **1990** | -0.02 | 0.07 | -0.07 | 0.27 | 0.04 | 0.28 | 27 | Excellent |
| **2001** | 0.14 | 0.19 | 0.42 | 0.59 | 0.28 | 0.38 | 104 | Good |
| **2010** | -0.05 | 0.15 | -0.10 | 0.34 | 0.05 | 0.37 | 268 | Good |
| **OA** | | | | | | | | |
| **1990** | 0.01 | 0.47 | 0.01 | 0.26 | 0.07 | 0.26 | 27 | Excellent |
| **2001** | -0.21 | 0.60 | -0.09 | 0.25 | -0.01 | 0.27 | 103 | Excellent |
| **2010** | 0.20 | 0.50 | 0.09 | 0.23 | 0.09 | 0.23 | 269 | Excellent |
| **Sulfate** | | | | | | | | |
| **1990** | 0.13 | 0.23 | 0.09 | 0.15 | 0.20 | 0.23 | 27 | Excellent |
| **2001** | 0.19 | 0.40 | 0.14 | 0.29 | 0.31 | 0.38 | 101 | Average |
| **2010** | 0.07 | 0.31 | 0.04 | 0.17 | 0.16 | 0.25 | 287 | Excellent |
| **Nitrate** | | | | | | | | |
| **1990** | 0.01 | 0.18 | 0.03 | 0.65 | -0.31 | 0.59 | 27 | Average |
| **2001** | -0.15 | 0.32 | -0.14 | 0.30 | -0.19 | 0.47 | 97 | Good |
| **2010** | -0.25 | 0.32 | -0.26 | 0.34 | -0.34 | 0.49 | 282 | Average |
| **Ammonium** | | | | | | | | |
| **1990** | -0.04 | 0.14 | -0.06 | 0.22 | 0.07 | 0.24 | 27 | Excellent |
| **2001** | 0.02 | 0.20 | 0.02 | 0.21 | 0.10 | 0.26 | 96 | Excellent |
| **2010** | 0.09 | 0.16 | 0.11 | 0.20 | 0.19 | 0.26 | 286 | Excellent |
| **PM$_{2.5}$** | | | | | | | | |
| **1990** | 1.20 | 1.63 | 0.23 | 0.31 | 0.19 | 0.26 | 27 | Excellent |
| **2001** | 1.49 | 2.72 | 0.12 | 0.23 | 0.09 | 0.23 | 951 | Excellent |
| **2010** | 1.10 | 2.08 | 0.12 | 0.22 | 0.09 | 0.23 | 938 | Excellent |










**Table 3**: Average observed and predicted PM percentage changes, and Pearson's
correlation coefficient calculated for each comparison case.

| | Observed changes (%) | Predicted changes (%) | Pearson's r | Number of Sites |
|---|---|---|---|---|
| **EC** | | | | |
| **1990 to 2001** | -19 | -12 | 0.80 | 21 |
| **2001 to 2010** | -19 | -17 | 0.39 | 75[a] |
| **1990 to 2010** | -45 | -24 | **0.91**[b] | 21 |
| **OA** | | | | |
| **1990 to 2001** | -16 | -13 | 0.68 | 21 |
| **2001 to 2010** | -18 | -9 | -0.32 | 89 |
| **1990 to 2010** | -33 | -23 | -0.16 | 21 |
| **Sulfate** | | | | |
| **1990 to 2001** | -9 | -9 | **0.88** | 21 |
| **2001 to 2010** | -35 | -22 | **0.92** | 75 |
| **1990 to 2010** | -40 | -29 | **0.97** | 21 |
| **PM$_{2.5}$** | | | | |
| **1990 to 2001** | -10 | -14 | **0.81** | 21 |
| **2001 to 2010** | -21 | -20 | **0.61** | 636 |
| **1990 to 2010** | -28 | -30 | **0.82** | 21 |



[a] 14 CSN sites reporting increases of EC, probably due to the change in the
measurement protocol in the 2007-09 period, have been excluded from this analysis.

[b] The correlations in bold are statistically significant for a significance level of 5%.








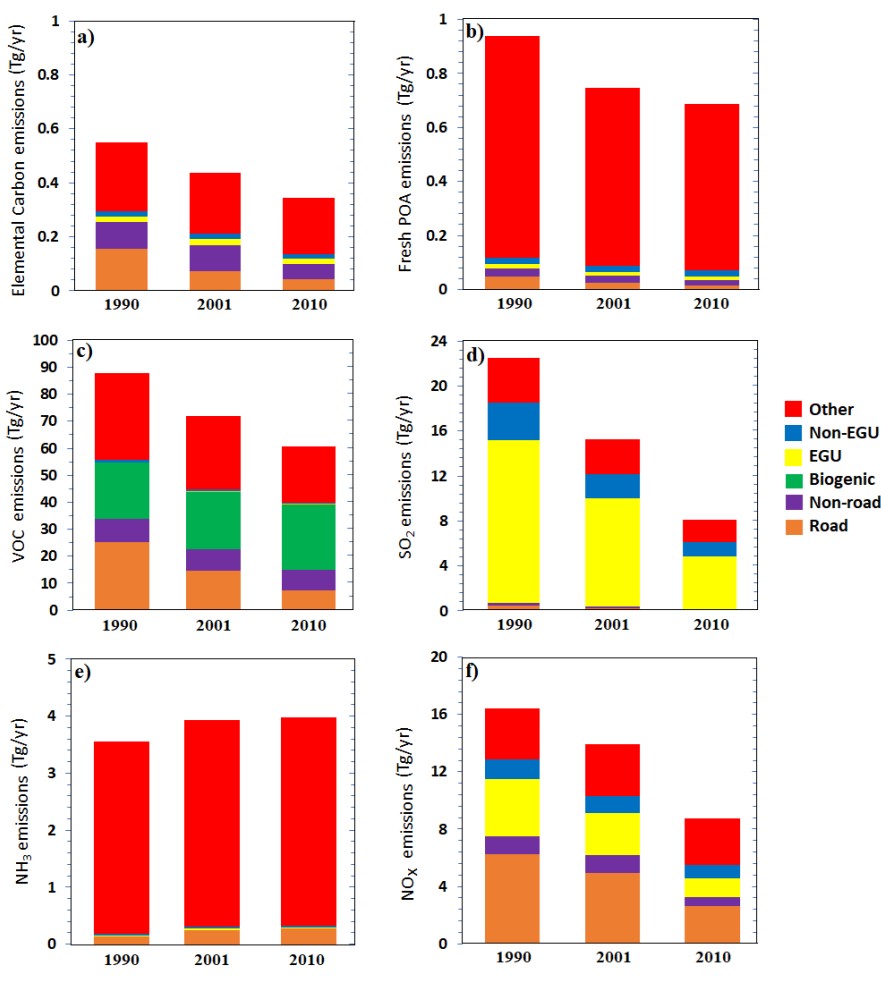

**Figure 1**: Annual emissions by each source for the whole domain for: a) elemental
carbon, b) fresh POA, c) non-methane VOCs, d) $SO_2$, e) $NH_3$, and f) $NO_x$.






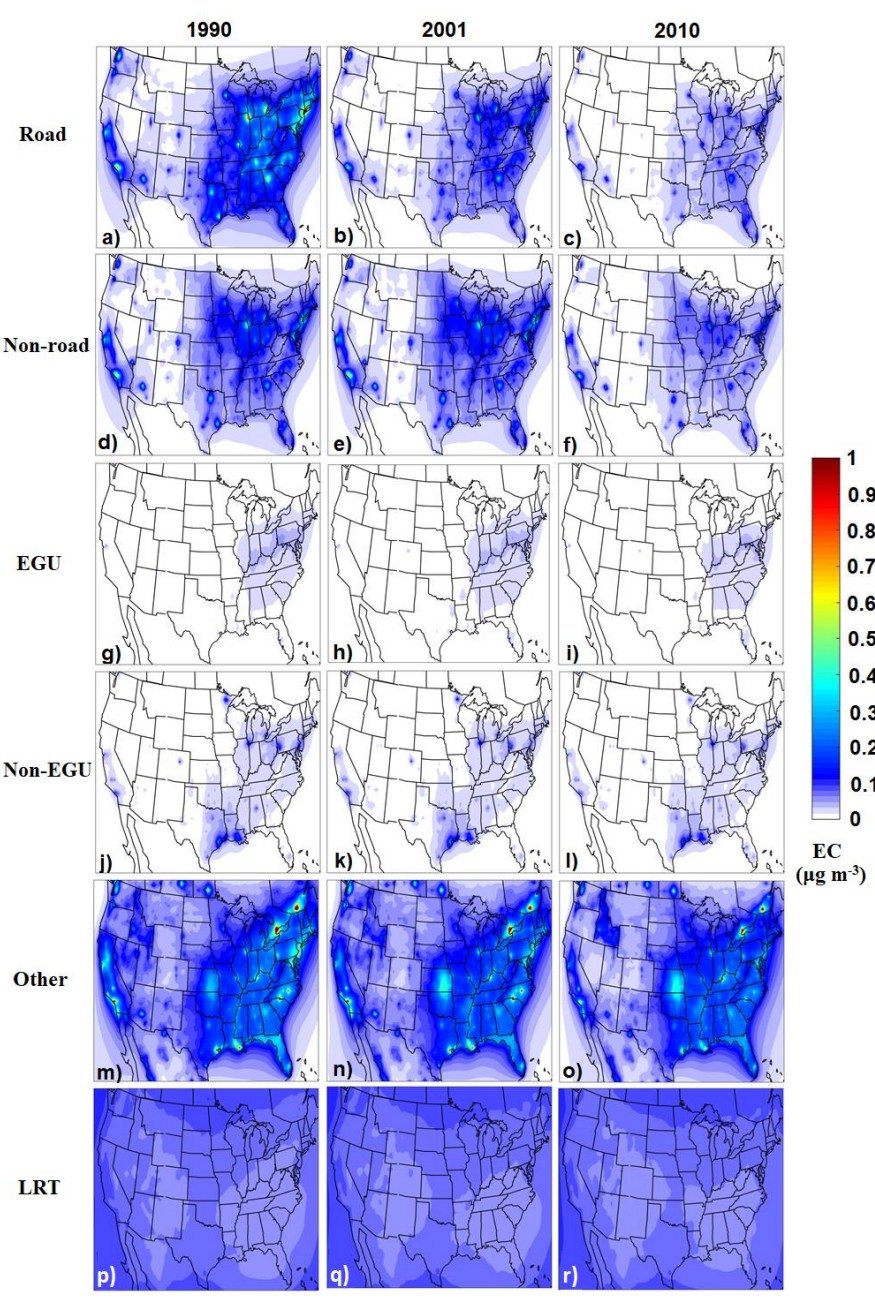

**Figure 2:** Predicted annual average ground level PM$_{2.5}$ elemental carbon concentrations per source for 1990, 2001, and 2010.

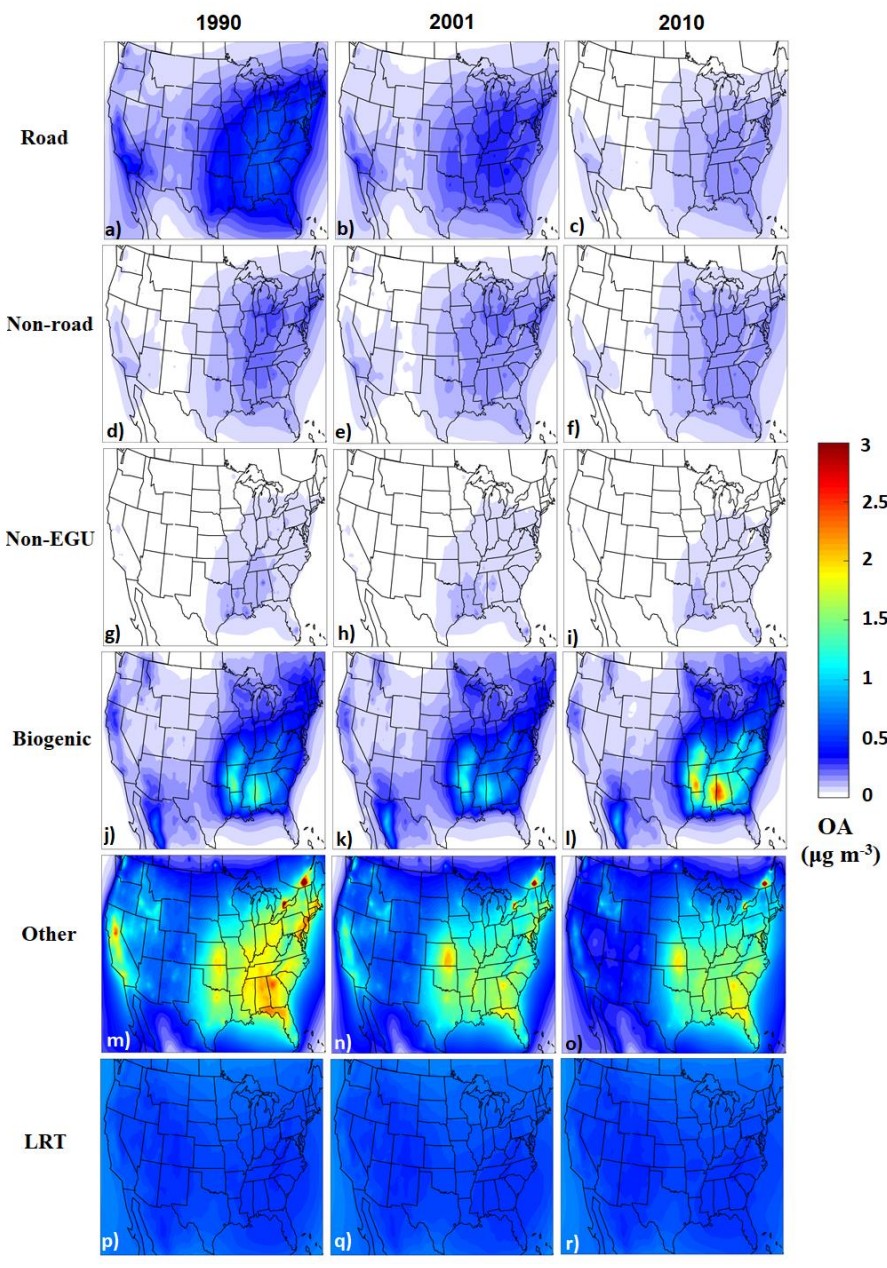

**Figure 3:** Predicted annual average ground level PM$_{2.5}$ organic (primary plus secondary) aerosol concentrations per source for 1990, 2001, and 2010. The EGU contributions are low and are not shown.





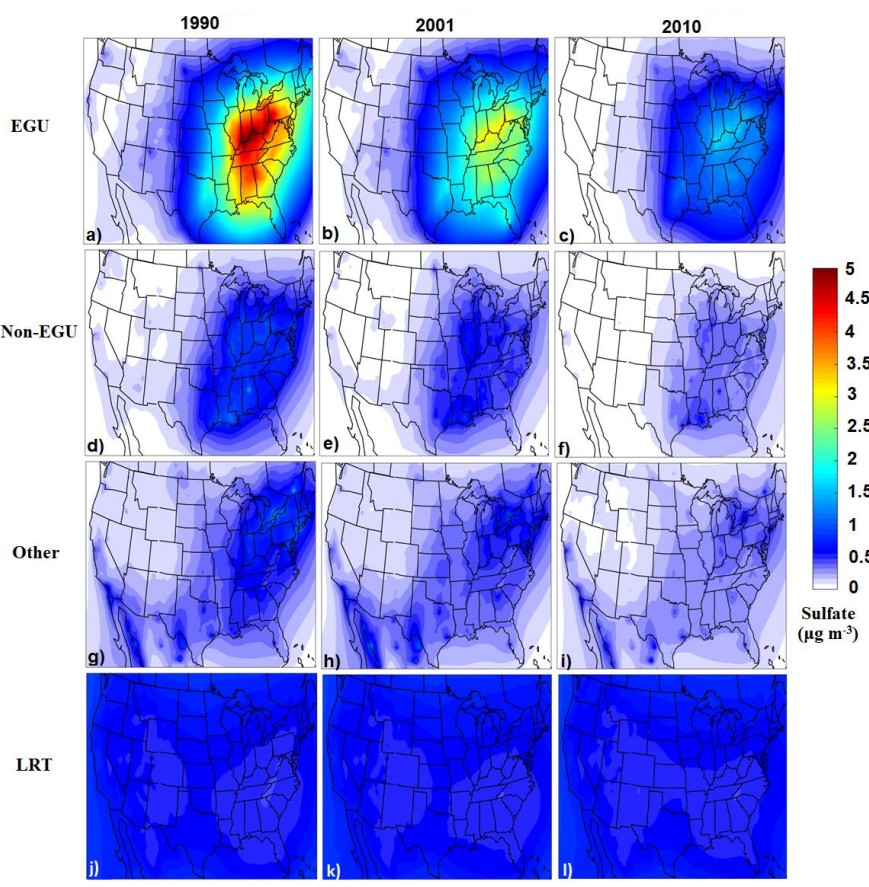

**Figure 4:** Predicted annual average ground level PM$_{2.5}$ sulfate concentrations per source for 1990, 2001, and 2010. The on-road, non-road, and biogenic contributions are low and are not shown.

















**Figure 5:** Definition of the 7 regions used in the analysis.






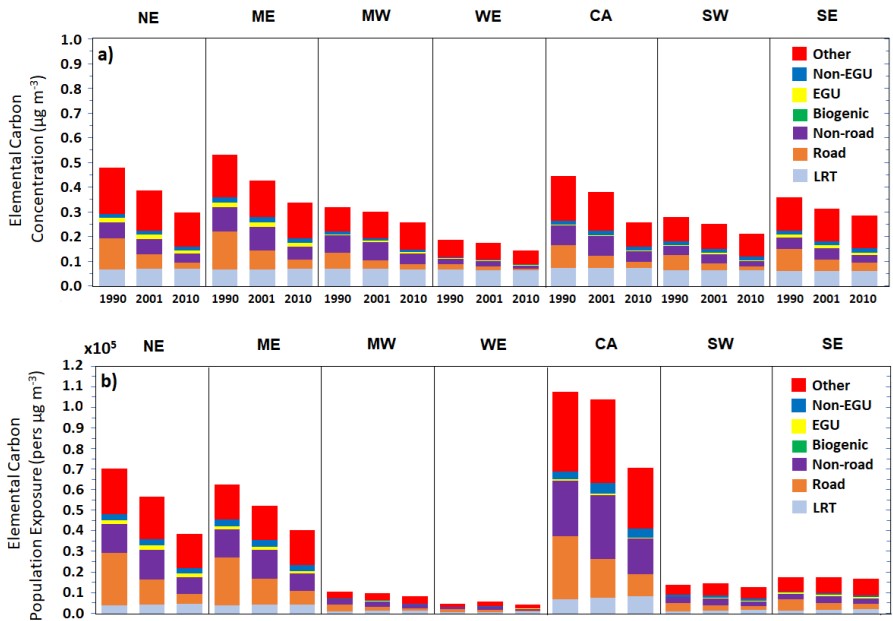




**Figure 6:** Sources of PM$_{2.5}$ EC for the different regions during 1990, 2001, and 2010 for: a) average concentrations (µg m$^{-3}$) and b) population exposure (persons µg m$^{-3}$).






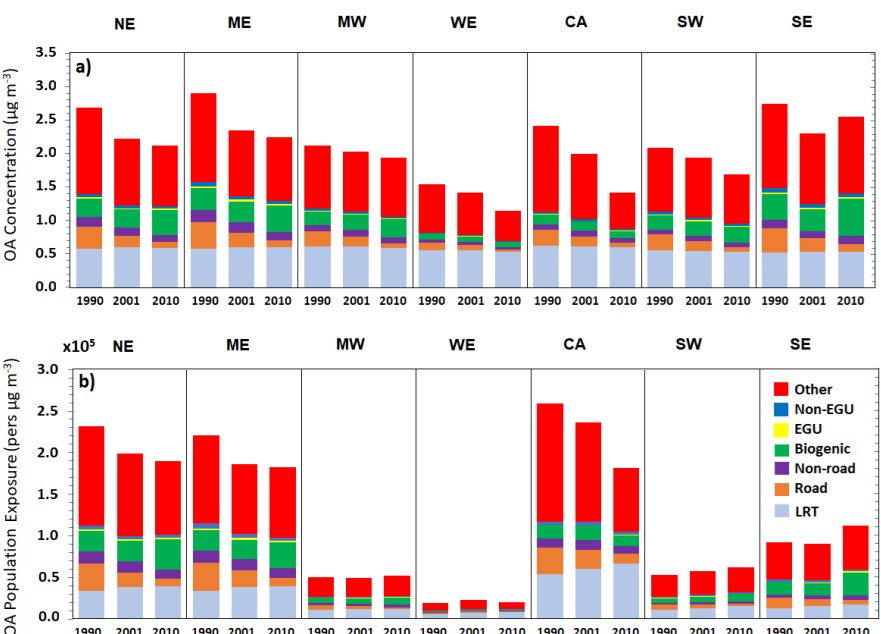




**Figure 7**: Sources of PM$_{2.5}$ OA for the different regions during 1990, 2001, and 2010
for: a) average concentrations (μg m$^{-3}$) and b) population exposure (persons μg m$^{-3}$).





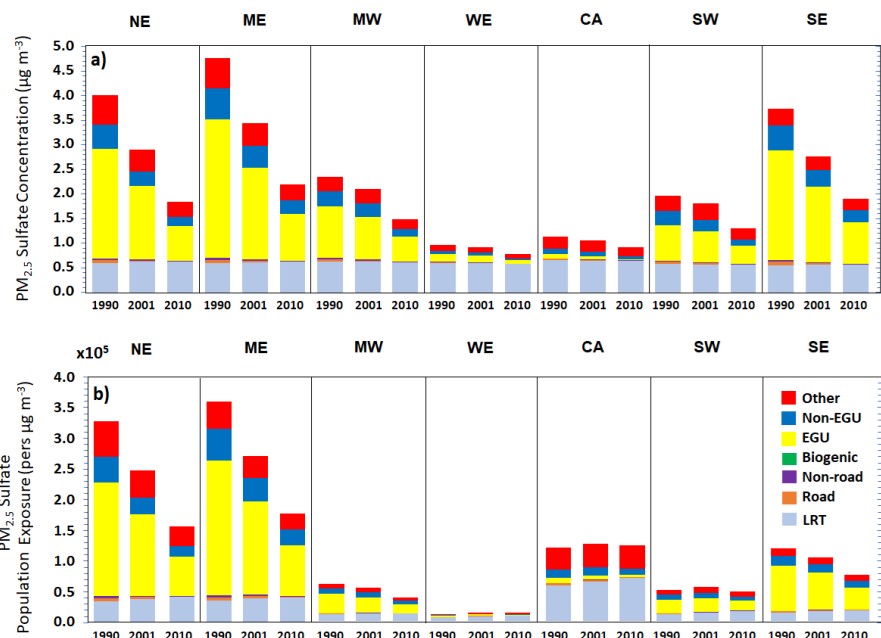

**Figure 8**: Sources of PM$_{2.5}$ sulfate for the different regions during 1990, 2001, and 2010 for: a) average concentrations (μg m$^{-3}$) and b) population exposure (persons μg m$^{-3}$).

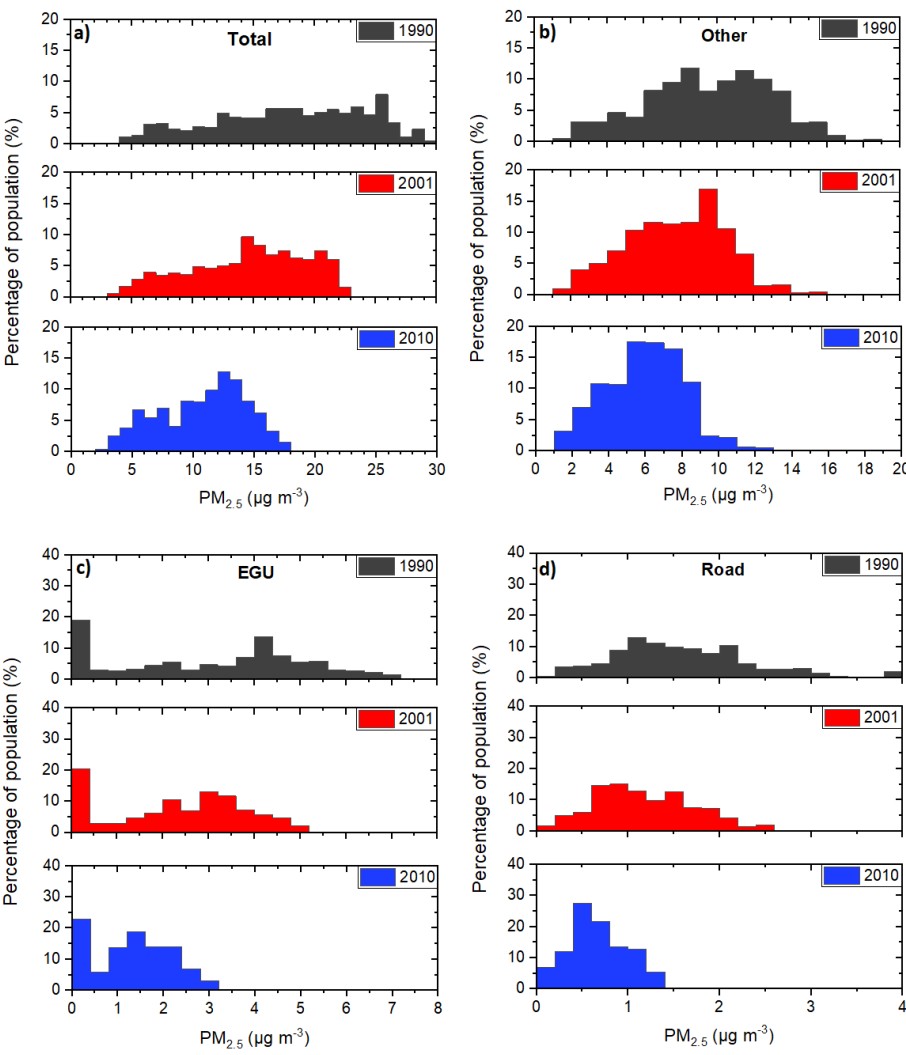

**Figure 9:** Distributions of population exposed to annual average PM$_{2.5}$ during 1990 (grey), 2001 (red), 2010 (blue); and for the dominant sources of PM$_{2.5}$: a) road transport, b) EGU, c) other, and h) total PM$_{2.5}$.