# Peer review of "Changes in PM2.5 concentrations and their sources in the US"

_Atmospheric Chemistry and Physics, 2021_

## Author Response (AR1)

**Responses to the Comments of Reviewer 1**

**(1)** What do authors mean by exposure and how is it estimated differently when compared to concentration? Is the exposure estimated considering the variable populations in 1990, 2001 and 2010 or was the population kept fixed?

We define exposure here following Walker et al. (1999) as the product of the average concentration times the population in each grid cell. The population distribution is based on the US census bureau data (www.census.gov/) and is different for 1990, 2001 and 2010. The population distribution of 2001 is assumed to be the same with that of 2000. These clarifications have been added to the revised paper.

**(2)** The authors represent 2 decades 1990-2010 with 3 representative years: 1990, 2001, 2010. Are 3 individual years enough to give a complete picture of the effect of change in emissions on concentrations? This question becomes even more important when authors estimate the exposure and hence authors should also include observed vs predicted exposure in grids where observations are available in Table 3.

The changes in concentrations during these years are due to both changes in emissions and also to the year-to-year variability of the concentrations due to meteorology. We have chosen these years that are a decade from each other because in almost all cases the effect of the significant changes in emissions is expected to dominate the year to year to variability. This is consistent with the analysis the emissions used by Xing et al. (2013). We do not argue that each year is mathematically representative of the corresponding decade (e.g., 1990 of 1985-1995). They should be viewed as three snapshots of US air quality in time that reflect mostly changes in emissions plus some year to year meteorological variability. Please note that because of the way that exposure is defined (concentration times population) and the population is measured, the evaluation metrics of our exposure predictions are exactly the same as the evaluation metrics of our concentration predictions. We have added the above discussion to the paper.

**(3)** The SOA exposure is in single digits in Table 3 which is much smaller when compared to concentrations. What is the reason behind the same?

The referee probably refers to the changes in exposure in Table 1, given that Table 3 presents predicted and observed changes in concentrations. The relationship between the changes in concentrations and exposure is strongly affected by the corresponding distributions in space. The more similar these distributions are, the closer to each other the corresponding changes. For example, for the SOA from road traffic the SOA concentration was reduced by 71% from 1990 to 2010 and the exposure was reduced by 66%. On the other hand, for non-road sources with a lot of them located in low population density agricultural areas, the corresponding SOA concentrations (according to PMCAMx) were reduced by 17%, but the exposure by only 6%. A significant part of the corresponding SOA reductions took place in areas with only a few people. The explanation of this important point has been added to the discussion of the results.

**(4)** Population plays an important role when estimating exposure; viz: higher population will result in greater no people being exposed to the same concentration when compared to smaller population. Since population increased during 1990-2010 in USA, reduction in pollutant emissions should be high enough so as to negate the factor of increased population in order to indicate an overall reduction in emission of a pollutant. Is it the same case here?

This is a good point and indeed this is the case here. The US population increased by 23.7% from 1990 to 2010 and if the emissions and concentrations had remained constant, the total exposure would have increased by the same percentage. The fact that the exposure to PM was reduced in most areas indicates that the reductions in concentrations were sufficient in most areas to overcome this effect and still lead to significant overall population exposure reductions. This point is now made both in the abstract and in the conclusions of the paper.

**Responses to the Comments of Reviewer 2**

**(1)** The main concern has to do with evaluation and how it is done and how it is characterized. First, the performance criteria provided are taken from Morris et al. (2005). Those criteria are for assessing daily simulations, not annually-averaged concentrations. One expects much better error statistics (annual biases will not be affected as such). The authors should go back and do a daily evaluation for the applicable periods modeled. This is really a must. Also, there has been updates by the Ramboll team in terms of performance evaluation (e.g., Emery et al., 2017, doi.org/10.1080/10962247.2016.1265027) which should be used. They also make recommendations on considering subdomains in the evaluation. A second major issue is that they do not include California in their analysis, stating: "We have excluded the region of California from this analysis because the coarse resolution used in this application does not allow PMCAMx to capture the significant gradients and high concentrations observed in that area." If that is the case, it cannot capture the exposures properly, either, and thus that area should not be included in the analyses. Either California should be included in the evaluation, or it should not be included in the rest of the analyses. Following the recommendations of Emery, their result would suggest California results should be subject to an independent evaluation. Also, the evaluation typically precedes the rest of the results given its importance.

We have followed the suggestion of the reviewer and added the results of the daily evaluation to the revised manuscript. These are now shown in Table 3 of the revised paper. We have also included the California region in the evaluation metrics presented in the main manuscript and the same metrics without California in the Supplementary Information. We have added the three additional metrics suggested by Emery et al. (2017) both for the daily and the annual average concentrations in two new tables in the Supplementary Information section for completeness. Finally, we have changed the order of the presentation of the results and now the evaluation is presented right after the description of the concentration fields.

**(2)** A second concern it their statement "The predicted reductions in sulfate concentrations are less than the reductions in emissions due mainly to the non-linearity of the aqueous-phase conversion of $SO_2$ to sulfate (Seinfeld and Pandis 2016) (Table 1)." While it is true that the formation of sulfate is somewhat non-linear, they have not shown this to be the major reason. This is highlighted by the difference between the reduction in concentrations and exposures, which, all else equal, must be linear given how it is calculated. They state that the difference between the concentration reduction and the exposure reduction as "The change in exposure is a little less, -40% on average due mainly to the location of the major $SO_2$ sources relatively away from urban centers." I think they mean that where the reductions occur are mainly away from the urban centers (this is an important difference: if the reductions were uniform in a relative sense, this difference should not occur). Might that also be important in terms of average levels? i.e., if the reductions occur in areas where the conversion is slower, the chemistry can be linear and one gets the result found here. Also, the sentence starting on line 364 is awkward.

This is a valid point. We do not include a detailed analysis of the causes of the non-linear response of the sulfate concentrations to the emission reductions because it would make the paper even more complex. We have rephrased this sentence just stating that such non-linearity has been predicted in past CTM applications and was mostly due to the non-linearity of the aqueous-phase chemistry. We have provided the corresponding references for this point.

We also agree that the second point needs clarification. We now explain that both the major sources of $SO_2$ and the higher reductions of sulfate are located and take place, according to PMCAMx, away from the major urban centers. We have rewritten this rather confusing sentence.

**(3)** The authors have "The major reasons for this behavior are the simultaneous decreases in $NO_x$ that have led to increased SOA formation yields and the time required for the formation of this SOA which is often produced away from its sources in high urban density areas. The reasons for this behavior are complex and will be analyzed in detail in future work. However, at least part of the explanation is due to decreases in $NO_x$ concentrations over the same period and associated increases in SOA yields." The second sentence seems to suggest that the first sentence is incomplete and will be assessed in the future. This is awkward.
We have rewritten these three sentences deleting the third one and rephrasing the first two.

**(4)** In calculating exposure, the authors should use the census for the years modeled. Population distributions change over time. Maybe have both results (stagnant population and a dynamic one).
This is exactly what we have done. The population distribution is based on the US census bureau data (www.census.gov/) and is different for 1990, 2001 and 2010. This clarification has been added to the revised paper. This point was also made by Reviewer 1 (comment 1). We have also added a brief discussion about the 23.7% increase in population in the US from 1990 to 2010 which would have led to a corresponding increase in total population exposure if the emissions/concentrations had not changed during the period studied.

**(5)** The time period modeled is getting rather old. Why not include 2020?
Unfortunately, the US emissions inventory for 2020 is not currently available. A major strength of the Xing et al. (2013) inventories used in this work is that they were prepared using identical methodology. Adding a rough inventory for 2020 based on extrapolations and limited data would weaken the overall effort. Our intention is to repeat this exercise for 2020 as soon as the corresponding inventory is available.

**(6)** More discussion on how this work compares to other studies that have examined the impact of emissions changes on air quality is in order (e.g., the EPA group, others). Are there any significant differences in their findings? If not, what is the main scientific contribution?
We have followed the reviewer's suggestion and added discussion to the previous efforts to examine the impact of changes in emissions in air quality. The major contribution of our work is

not so much the link of emission changes with concentration changes but rather the source-resolution of our analysis. This is now stated clearly in the abstract and the conclusions.

**(7)** *The "," in line 330 is not needed.*
Corrected.

---

## Author Response (AR2)

**Responses to the comments of the reviewer**

**(1)** The manuscript has been improved, and the authors have addressed many of the issues identified.

We appreciate the positive assessment.

**(2)** The evaluation is still mischaracterizing how they classify performance. The Ramboll work should not be used to say that the annual performance is excellent/good/average (They state: "Based on these criteria, the ability of the model to reproduce the annual average concentrations of the sites is excellent for OA, good to excellent for PM2.5, EC, and ammonium, good for sulfate, and average for nitrate." The Emery et al. metrics were developed purely for assessing a model's ability to simulate daily quantitates. As the authors clearly show, one might expect much better performance using annual values, but that also can cover up major problems (e.g., large, but off-setting biases by season). This paragraph must be changed. They can present the specific values, but should not provide an evaluation of how good they are unless they have a scientific basis for such, e.g., a peer-reviewed paper or agency report that has laid out an evaluation scale for annual performance.

We have followed the suggestion of the reviewer and removed the characterizations (excellent, good, average, problematic) of the annual performance of the model both from the text and Table 1. We have also removed the corresponding column from Table S2 in the Supplementary Information of the paper. These characterizations are only used now for the daily performance. The performance metrics are presented for both the annual and daily averages.